# Development of the Refugees and Asylum Seekers Occupational Satisfaction (RASOS) Assessment Tool

**DOI:** 10.3390/ijerph20196826

**Published:** 2023-09-25

**Authors:** Pavlina Psychouli, Ioulia Louta, Constantina Christodoulou

**Affiliations:** Department of Health Sciences, European University Cyprus, 2404 Nicosia, Cyprus; i.louta@external.euc.ac.cy (I.L.); c.christodoulou@external.euc.ac.cy (C.C.)

**Keywords:** refugees, asylum seekers, assessment tools, barriers, occupational needs

## Abstract

The objective of this study was to develop an occupation-centered and client-centered assessment tool for refugees and asylum seekers. A preliminary tool outline was produced based on a literature review, while considering previous published tools’ strengths and limitations. A qualitative study was undertaken via focus groups to improve on the tool’s design and adequacy for its purpose, resulting in the creation of a pilot version of the tool. Convenience sampling included 8 Greek and Cypriot professional and student occupational therapists with experience in the field, 8 international expert occupational therapists, 4 laypeople, 4 humanitarian professionals, and 5 refugees and asylum seekers. Basic qualitative content and thematic analysis led to topics regarding tool modifications that concerned categorization, formation/structure, wording, administration, and assessment scale. Corresponding tool revisions ensued. This study led to the development of the pilot version of the Refugees and Asylum Seekers Occupational Satisfaction (RASOS), which can also be used to identify underlying personal and environmental factors that contribute to self-perceived low satisfaction. A future quantitative study is required to establish the psychometric properties of the tool.

## 1. Introduction

Forced displacement occurs when individuals forcibly leave their homes to avoid armed conflict, generalized violence, human rights abuses, natural or man-made disasters, and/or development projects [1]. Refugees are people who flee their country because of a well-founded fear of persecution due to nationality, race, religion, a specific social group membership, or a political opinion and qualify for international protection under the 1951 Refugee Convention and its 1967 Protocol [2]. Those awaiting to receive a decision on their asylum claim are called asylum seekers [3].

Refugees and asylum seekers face many psycho-social difficulties associated with resettlement [4], which relate to personal, social, and cultural factors [5]. The disruption of rituals, routines, and roles, in conjunction with repetitive trauma and instability [6], interferes with participation in meaningful occupations, referring to what we call “occupational injustice” [7,8]. Occupational injustice is experienced when a person is denied of their occupational rights and is marginalized from or deprived of the opportunity to engage in meaningful occupations [8].

A literature review demonstrates that there is a broad range of occupations that displaced persons are excluded from, thus limiting occupational participation, performance, and satisfaction [9]. These often relate to meaningful leisure, social, and work occupations. This means that resettled refugees have limited access to healthcare services and employment and face linguistic, social, as well as financial isolation [10]. For those residing in detention centers, the negative impact on mental health, occupational performance, and quality of life is far greater, affecting participation and satisfaction in even the most basic of daily activities [11].

Human life is characterized by carrying out everyday activities, within a temporal, physical, and sociocultural context [12]. Occupational therapy acknowledges the association between participation in occupation and health [13], which illustrates the necessity to directly address occupation with displaced persons [14], through occupation-centered assessment [15].

As a complex construct, it is improbable to universally define occupation [16], which needs to be politically [17,18] but also culturally [19], temporally, and ecologically contextualized [16]. In Western cultures, occupations are defined as “the everyday activities that people do as individuals, in families, and with communities to occupy time and bring meaning and purpose to life” [20]. In the African [10] and the Eastern culture, individuals’ occupations are more intertwined with the occupations of others [21]. Thus, even though occupations are universal, occupational significance, or meaning, may vary across cultures [22]. Meaning is how people make sense of their occupational performance [23] and leads to the understanding of the individual and collective needs and goals of people [24]. As experiences meaningful to the individual are strongly linked with occupational satisfaction [25,26], a focus on personal perspectives of satisfaction [26] emerges as a key characteristic of occupation-centered assessment.

To this day, a review of the literature reveals that non-occupation-specific instruments have been used by researchers with a focus on well-being and quality of life [27,28,29,30,31], such as the World Health Organization Quality of Life-100 (WHOQOL) [32] and the Personal Well-Being Index (PWI) [33]. Needs assessment tools, such as the Camberwell Assessment of Need [34], and assessments related to psychiatric disorders, and especially post-traumatic stress disorder [35], have been utilized. These lack a much-needed focus on occupation.

Common occupation-centered tools have been used to identify the displaced persons’ perspectives of their performance difficulties [27,36,37,38]. The Canadian Occupational Performance Measure (COPM) has been used to capture the change in performance and satisfaction in all areas of life (self-care, leisure, and productivity) of Syrians under temporary protection in Turkey [36]. The COPM was opted for as it is fast to complete, which makes it attractive for refugees and asylum seekers settings, where time is usually lacking. Even though a translated version of the COPM was used (in Turkish), along with the translation to Arabic, the COPM remains an assessment that is rooted in the Euro-Western culture [38]. The same applies for the Activity Card Sort (ACS) and the Satisfaction with Daily Occupations (SDO) tools, which were used in another study [27] to assess asylum seekers’ satisfaction with daily occupations and activity level while in a Danish asylum center. In the same study, the Assessment of Motor and Process Skills (AMPS) was used to assess performance capacity for activities of daily living and/or independent living. The AMPS has been cross-culturally standardized but, as it measures none other than motor planning and cognitive function, it lacks the potential of an exhaustive and holistic assessment of occupations and offers limited information on the actual occupational performance and the degree to which the individual is satisfied by this.

The use of standardized assessments proves to be limited, possibly due to a lack of culturally aware assessments [37]. One of the main tools used in occupational therapy assessment and intervention is the Occupational Therapy Practice Framework (OTPF) [39], which describes the central concepts and assumptions related to everyday activity, how it can be categorized, environments in which it may take place, and factors that may promote or hinder occupational engagement. OTPF could serve as a guide toward development a culturally aware tool for displaced populations but has not been utilized in this way, up until today.

Therefore, taking into consideration the importance of occupation-related assessment as the first step in the facilitation of psychosocial support for integration and the lack of such population-specific, activities-centered assessment tools, the purpose of this study was to design a client-centered tool that would be specifically designed for refugees and asylum seekers and would provide insight to the following:The degree of the forcibly displaced persons’ satisfaction with their occupational participation in the host country;Potential factors that may be related to a reported lack of satisfaction;Significant occupations in which the people themselves desire to participate;Occupations they do not have access to and the reasons for this.

## 2. Materials and Methods

### 2.1. Research Design

This was a qualitative study, aiming to develop a pilot assessment tool. The study took place in Cyprus and was designed by taking into consideration the context, setting, and participants’ frame of reference [40]. The research design included two consecutive phases: first, a literature review, and secondly, focus groups and reviews, interchangeably.

### 2.2. Ethical Issues

Participants were handed out an information sheet explaining the purpose and methodology of the study and signed a consent form before being included in the study. Ethical approval was granted by the Cyprus National Bioethics Committee (reference number ΕΕΒΚ ΕΠ 2020.01.27).

### 2.3. Participants

Convenience sampling was used in the study. Thirty-one individuals were invited to participate. Recruitment was implemented by Authors 2 and 3. Inclusion criteria for each of the study’s phases were as follows:(a)Focus Group I

Participants were recruited from the European University Cyprus (EUC) through phone calls and emails, between January 2020 and March 2020. Participants had to be professional occupational therapists and occupational therapy students, with at least two months’ experience working with displaced individuals, fluent in English, and citizens of Cyprus or Greece.

(b)Focus Group 2

Participants were recruited through social media posts between March 2020 and May 2020. The aim was to include a sample with as wide a variance as possible in terms of countries in which the relative working experience had been gathered.

Participants had to be international occupational therapists with at least 1 year of experience working with displaced individuals and fluent in English.

(c)Review 1

Participants were recruited through phone calls between May 2020 and July 2020. The aim was to include people that were neither occupational therapists nor other experts in the field. Participants had to be laypeople and fluent in English.

(d)Review 2

All participants were recruited through emails and phone calls between November 2021 and January 2022. Participants had to be health or humanitarian professionals, other than occupational therapists; residents of Cyprus and working with displaced persons at the time, or with past relative experience of more than 1 year; and fluent in English.

(e)Focus Group 3

Participants were recruited through email and phone calls between January 2022 and March 2022. The aim was to include a sample representative of the population of interest, varying in terms of status, age, gender, and country of origin. Participants had to be residents of Cyprus at the time and fluent in English.

### 2.4. Instruments

Focus groups were used as the main instrument to develop the preliminary draft tool, check tentative conclusions about tool items and topics about modifications, and identify unanticipated issues in an interactive and natural manner [41]. Reviews took place to detect possible language and outlook errors in the draft versions of the tool and to ensure they were understood and answered by respondents.

### 2.5. Procedures and Data Collection

The research procedure, which lasted from 2020 to 2022, included two phases. Phase 1—Establishment of the theoretical background/Creation of first draft tool (October–January 2020)—involved the initial literature review on evaluation tools regarding the population of interest and was conducted by Author 3 and reviewed by the other two. Phase 2—Further development of the tool (January 2020–March 2022)—involved two focus groups, followed by two reviews and a final focus group, all of which were implemented by Authors 1 and 3. Corresponding tool revisions were implemented by all authors. All respondents’ remarks were noted and reflected upon [42], resulting in the pilot version of the tool. An interview guide with key questions was created for focus groups. Preliminary interview guide questions were based on the literature concerning occupational participation and satisfaction of refugees and asylum seekers. The list was reviewed and expanded to include broader issues.

Focus group 1 (March 2020) was conducted at the EUC premises to collect data from occupational therapists with experience in the field. Participants received the initial draft version of the tool along with other assessment tools that had been used with the population under study in the literature. They were instructed to review the tools to form an idea that would help discuss the content and design the tool should have. Focus group 1 was audio-recorded. An interview guide with semi-structured questions helped to navigate discussion. Topics covered included categorization, formatting, wording, administration, and evaluation scale. Facilitators asked follow-up questions to clarify responses or prompt participants to elaborate on their statements. Tool revisions followed (March 2020).

Focus group 2 (May 2020) was conducted online due to COVID-19 restrictions and was video-recorded to collect more variant data across countries from occupational therapists with experience in the field. Participants were sent an online copy of the updated draft version of the tool with instructions to review it to be able to critically discuss the outlook, content, and procedure and raise potential topics or concerns. An interview guide was used with semi-structured questions for establishing that the tool measures what it is supposed to. Feedback received from Focus group 2 resulted in further changes. The draft tool was again revised (May 2020).

Review 1 (June 2020) involved one-on-one interviews with laypeople to ensure that the tool’s components and structure were clear and easy to understand by everyone, followed by tool revisions (July 2020).

Review 2 (November 2021) involved individual reviews on the draft instrument. Health and humanitarian professionals received the newest version of the tool and were asked to reflect on whether the tool was easy to understand and used by experts other than occupational therapists. Feedback was analyzed, leading to further tool changes (January 2022).

Focus group 3 (February 2022) was conducted online and was video-recorded. Participants were asked to provide feedback about usability, understandability, and relativity, critically analyzing the draft tool and posing concerns, which provided insight by the population of interest. An interview guide was used with semi-structured questions. The additional information was analyzed and processed, and the final version of the pilot tool was created (March 2022). Table 1 illustrates the methodologies employed for the focus groups and reviews.

### 2.6. Data Analysis

Data analysis was performed by Authors 1 and 3. Author 2 reviewed preliminary findings and helped check interpretations against the data for accuracy. Participants were asked to comment on the process to acknowledge and address any potential biases. Authors 1 and 3 took down notes on researchers’ thoughts and participants’ comments during focus groups and reviews. Authors 1, 2, and 3 used critical self-reflection to address subjective biases that might affect observations and findings. The audio recordings were transcribed verbatim. The transcripts were verified for accuracy by cross-checking them with the written notes taken separately by Authors 1 and 3. Basic qualitative content and thematic analysis were used. Open coding was conducted manually by Authors 1 and 3, separately. Author 2 coded the data and compared coding results with Authors 1 and 3. There was an intercoder agreement of 80% among Authors 1, 2, and 3. Themes of recurring patterns were constructed and sorted, resulting in topics that reflected what was observed in the data and dimensions of participants’ statements [41].

## 3. Results

### 3.1. Phase 1: Establishment of the Theoretical Background/Creation of First Draft Tool

The literature review led to the conclusion that there is a lack of occupation-centered and client-centered assessment tools especially designed for the needs of refugees and asylum seekers during resettlement in the host country. For the creation of a preliminary tool, the OTPF [39] was used to clarify and organize central concepts and adopt terminology, combined with the use of valuable constructs of pre-existing tools. The COPM [43] was used as a guide to develop a client-centered tool administration method in the form of a semi-structured interview. The WHOQOL [32] helped toward measuring occupational satisfaction and incorporating complex multidimensional concepts. The Model of Human Occupation and Screening Tool (MOHOST) [44] led to the addition of factors that affect satisfaction. Therefore, the first draft tool included an Introduction, Instructions, Part 1 (Demographics), and Parts 2 to 4 containing 34 evaluation components (occupations), with four columns each. Column A presented categories of occupations (involving basic needs and daily life occupations (Part 1); sleep/rest time and free/leisure time occupations (Part 2); and Work, education, and social participation (Part 3)) and provided two to three examples for each occupation. A five-point Likert Scale was used in Column B to measure each person’s satisfaction from participation in occupations. A zero (0) point was added as the “not-participating” choice. An additional column (C), with seven options, was designed to identify potential personal or environmental factors that might limit satisfaction, corresponding to an Annex. Column D was added for comments.

### 3.2. Phase 2: Further Development of the Tool

#### Participant Demographics

Table 2 illustrates the participants’ characteristics.

In Focus group 1, eight individuals participated: six professionals and two students of the EUC BSc Occupational Therapy Program. All had experience in the field, ranging from 3 months to 5 years. The age range was between 20 and 50 years.

In Focus group 2, eight international occupational therapists participated: one male lecturer from Belgium; two female researchers, one from Germany and one from Canada; two female clinical occupational therapists, one from Australia and one from the USA; one clinical occupational therapist from Greece; and two female EUC teaching staff. The experience of the participants in the field ranged between one and five years, while their age was between 25 and 50 years.

In Review 1, four individuals participated: two males and one female from Cyprus, and another female from Greece. The age range was between 20 and 50 years.

In Review 2, four individuals participated: two females, a social worker and a psychologist, and two males, a social advisor and a safety officer, all from Cyprus. The age range was between 30 and 55 years, while years of experience in the field ranged between 2 and 10 years.

In Focus group 3, five individuals participated: two male asylum seekers, one from Cameroon and another one from Central African Republic; a male refugee from Guinea; and two female refugees from Somalia. The age range was between 19 and 35 years.

### 3.3. Thematic Analysis

The topics that emerged after the focus groups and reviews regarding modifications proposed by participants concerned the (a) categorization, (b) formation/structure, (c) wording, (d) administration, and (e) assessment scale, leading to the pilot version of the tool (Table 3).

#### 3.3.1. Focus Group 1

Focus group 1 had a 140 min length.

Categorization: Tool items (occupations) were reorganized to clearly differentiate between categories of occupations.Formation/structure: Demographics was converted into table format to abbreviate the data collection method. An occupational profile was proposed and added to summarize the respondent’s past occupational experiences. The tool format was modified to include an Introduction, Part 1 (Demographics), Part 2 (Occupational profile), Instructions, and Sections A–C (Occupational items).Wording: An annex (Annex 1) was added with more examples of occupations, relative to the situation of refugees. The Instructions and Introduction were rephrased for clarity. More detailed information was provided about the initial annex, renamed to Annex 2 (Factors). Relevant examples to the state of displaced persons were added to Annex 2, one for each of the seven categories of factors. A glossary (Annex 3) was created to help clarify confusing scientific/occupational terms listed.Assessment scale: Column D (Comments) changed to “Are there any specific activities for which satisfaction might be different?” to explore this.

#### 3.3.2. Focus Group 2

Focus group 2 had a 90 min length and resulted in the following:Categorization: A new table was added. Ιt presented all occupation items. Respondents could select the three most urgent ones to address (Section D). Discussions concerned play, sleep, social participation, sexual activity, and categorization of items. Section A (Self-care, daily life at home and in the community, health management, and sleep/rest time), Section B (social participation, free time/play, and leisure activities), and Section C (Work and education) were reorganized.Formation/structure: The Instructions indicated the use of certain sections to curtail administration when necessary.Wording: The “not applicable” option of Column A (Categories of occupations) changed to “not assessed” for sensitive occupations. “Survival activities” were added to Annex 1 (Examples of occupations).Administration: Participants noted that administration at the initial reception phase during the asylum process might be challenging. The present tool would serve as a screening tool after the first response procedure.

#### 3.3.3. Review 1

The duration of each interview was 60–80 min.

Wording: The terms “occupation” and “satisfaction” were not clear to participants. Researchers added explanatory definitions in the Introduction.

#### 3.3.4. Review 2

Categorization: The occupation “educating/informing for legal rights and obligations” was divided to reflect these two diverse notions, adding one more evaluation component (35 in total).Wording: “Dialect” was added, along with “language”, in Demographics.Administration: The tool was described as useful for adolescents transitioning to adulthood. The tool administration age changed from >18 to >17 years old. Respondents’ fluency in English was stressed, to avoid misinterpretations. The phrase “be able to communicate effectively in the English language” was added in the Introduction. Administration with purposes other than screening, such as self-reflection, was recommended. The purpose of the tool remained the same.

#### 3.3.5. Focus Group 3

Focus group 3 had an 80 min length.

Wording: Column D (“Are there any specific activities for which satisfaction might be different?”) again changed to “Comments” as it confused participants. This time, it was clarified in the Instructions that both the interviewer and the respondent could comment. Further changes were made to the Introduction to refine phrasing of the term “Satisfaction”, which was mistakenly understood as the ability to engage in occupation.

## 4. Discussion

The research procedure of this study included a literature review and a qualitative study via focus groups and reviews that resulted in the design of a pilot assessment tool that measures satisfaction from participation in occupations and identifies the reasons that limit satisfaction in displaced populations.

The tool was created in English, a common language in refugee settings [45]. As discussed in Focus group 2, language barriers pose limitations in research and practice, while language forms an important part of culture. Having been designed with the intent to be administered by specialists who are fluent in English and can translate if the respondent is not also fluent, the tool would initially be in English and, later, translations in other languages could follow.

Reflecting on the use of the tool, one participant of Review 2 expressed the idea that apart from screening, the instrument could support self-reflective work “so that respondents could become aware of challenges and acknowledge the significance of change in their pursuit for safer, and more prosperous living conditions”. Participants in Focus group 2 also indicated that the tool could be used as a universal instrument when there is a lack of occupational therapists in emergency response and refugee settings. Also, in the spirit of collaborative practice by multidisciplinary teams of experts (occupational therapists, psychologists, social workers, or others), the tool could be used for developing effective crisis management strategies that reflect the desired occupational outcomes for integration [46]. 

With regard to understandability and usability, participants described the tool as clear and easy to use for the most part. In Review 2, one participant highlighted that it incorporates important components to “comprehensively understand the multidimensional nature of the multifaceted needs of a displaced person to integrate into the given society, as it addresses past and present experiences and evaluates internal and external affairs and concerns”. Expanding on this, Part 2 (Occupational profile) can have a catalytic role in providing a wealth of insights in the occupational history of respondents [47]. This would help to conceptualize changes in the individuals’ occupational satisfaction due to the experience of refugeeism, acknowledge personal meanings on occupations, and identify significant occupations for the respondent.

Participants agreed that occupations of Section A (Self-care, daily life at home and in the community, health management, and sleep/rest time), Section B (social participation, free time/play, and leisure activities), and Section C (Work and education) are all occupations refugees and asylum seekers are at times alienated from or deprived of in the host country. The level of importance of these varies and seems to be associated with social and cultural elements of satisfaction that need to be further researched in the future [22,26]. Prevalent and recurring concepts in discussions concerned establishing a culturally aware and sensitive approach to the development of the tool. Ensuring that administration would follow an ethical reasoning process that guarantees consent, respect, privacy, confidentiality, and practices that are meaningful, purposeful, and beneficial for the displaced persons was also highlighted. These add to the initial considerations about language barriers, resources, or trauma, emphasizing the importance of administering the tool after first response aid and by trained professionals in the field. Interviewers would make good use of the “not assessed” option in Column A (Categories of occupations) (Focus group 2, to avoid discussing irrelative or sensitive occupations due to personal experience, background, culture, or trauma [25], possibly rendering the tool even more suited for displaced persons.

At the same time, respondents can discuss the specific occupations they perceive as relative or urgent [48] as they are presented with a comprehensive cluster of occupations in Column A (Categories of occupations). This way, a client-centered tool is shaped that can be sensitive to individual and collective needs and situations. Particularly, assessing the level of satisfaction in Column B (Likert scale), or marking the absence of it, could acknowledge personal meanings on occupation [26]. This innovative feature of the tool could mean that it is more suitable for respondents with a diverse background, such as refugees and asylum seekers. Column C (Factors) can help to delve deeper into the issues of justice that are implied when self-perceived low satisfaction is noted, by exploring potential factors of occupational deprivation [7,49], which, as already highlighted, strongly concerns displaced individuals. Identifying and analyzing the factors that affect satisfaction from participation in certain occupations can help represent different perspectives on occupation and personal meanings relating to it, which may result from differences in the cultural or social background of respondents. For example, the satisfaction from participation in work or leisure activities may vary significantly across cultures and regions, which can be captured with the help of this column. Column D (Comments) can give a voice to possible concerns of both the interviewer and the respondent that might illuminate issues affecting the assessment process related to circumstances during tool administration, language use, or cross-cultural differences [50]. Further scrutinizing personal priorities in Section D can help to directly address the most urgent needs from the respondent’s point of view for sustenance during resettlement [25].

Annex 1 (Examples of occupations) can contribute to relativity, too. It includes examples of occupations that are related to the situation of refugees and asylum seekers based on real-world data gathered in clinical settings with displaced persons provided by the specialists with work experience in the field and displaced individuals during focus groups. According to participants in Focus group 1, aspects of personal hygiene and care were considered important additions as following culture norms for certain social instances is expected in Western societies [51]. Occupations regarding life at home are critical because of the vastly different living conditions in households of a high-income Western country [52]. Activities within the community and self-advocating are essential because they relate to independence and self-sufficiency [53]. Religious or spiritual occupations relate to personal resilience and expression of identity [54]. Managing communications with public services is often confusing for newly arrived individuals, particularly considering the language barrier [8]. Important issues often concern characteristic elements of life in a Western society, such as the management of personal documents and accounts [52]. As access to the labor market is very limited for asylum seekers [55] being informed about work norms and procedures is important for starting and maintaining a job position. Young adults might face additional challenges as they transition to adulthood in a strange country that is often unwelcoming [56] These could be associated with family planning; identifying interests, skills, and opportunities; and recognizing available systems for psychosocial support. Examples in Annex 2 (Factors) were also based on real-life data contributed by specialists and displaced persons during focus groups. Annex 2 is structured and phrased to also be able to contribute to relevance, as participants in Focus group 2 observed, by helping conceptualize factors that affect participation on a micro-, meso-, and macro-level [57] in a tangible manner that can resonate with displaced persons.

Findings of this study complement existing research on the occupational assessment of forcibly displaced persons. The development of the present tool has been an attempt to cover the gap identified in the literature in assessment tools used for populations such as refugees and asylum seekers.

The results suggest that an occupation-centered assessment focused on satisfaction can identify the specific occupational needs of refugees and asylum seekers during resettlement in the host country, as they perceive them. Implications of this study highlight the importance of further research of cross-cultural components pertaining to occupation and satisfaction related to language, culture, and significance variables [58]. It is needed to study how perceptions of occupational satisfaction can be used as an analytical tool to gain a better understanding of refugees and asylum seekers’ occupational needs in the host country and how practices such as occupation-centered and client-centered assessment can lead to emancipatory and transformative social actions.

Consequently, it is suggested that this pilot tool should be further tested for validity and reliability, allowing for the end assessment instrument to be used in the future as a standard evaluation method in refugee settings.

This study has several limitations. The generalizability of the results may potentially be limited by using a convenience sample that represented mostly one Western high-income country of resettlement (Cyprus) and had a level of fluency in the English language. Refugees and asylum seekers participated in Focus group 3, as it was necessary to first lay the scientific foundations of the tool. Unfortunately, displaced individuals only from the African continent ended up participating, representing a potential bias for the results of the study. Focus groups 2 and 3 were held in English. Non-native English-speaking participants might have faced difficulties comprehending facilitators’ questions and other participants’ comments, communicating their positions, and elaborating on their contributions. Another potential limitation is that data analysis relies solely on mutual reviews among the three authors. Future studies may use a cross-checking method, including additional researchers.

## 5. Conclusions

This study has led to the development of a pilot version of the RASOS instrument. RASOS is an occupation-centered and client-centered assessment tool for refugees and asylum seekers, which provides a framework for measuring occupational satisfaction from participation in occupations and helps identify underlying personal and environmental factors that contribute to self-perceived low satisfaction. The use of this tool in different countries and various populations can provide insightful data on significant occupations, occupational challenges, and attitudes toward participation across cultures and regions. This tool can also be used by different specialists to inform the needs assessment process of forcibly displaced persons during resettlement in a host country globally. Moreover, in the future, a quantitative study is required for pilot testing, the establishment of structural validity, and the estimation of internal consistency.

## Figures and Tables

**Table 1 ijerph-20-06826-t001:** Methodologies of focus groups and reviews.

	Focus Groups			Reviews	
	1	2	3	1	2
**Setting** **/** **location**	In person (EUC ^1^ premises)	Video Conference	Video Conference	In person (EUC premises)	Video Conference
**Purpose**	Discuss content and design	Review draft version, critically discuss outlook, content, and procedure	Discuss usability, understandability, and relativity	Discuss clarity and understandability of components and structure	Discuss ease of use by professionals other than occupational therapists
**Population**	Greek and Cypriot professional Occupational Therapists and Occupational Therapy students	International OTs	Refugees and asylum seekers residing in Cyprus	Laypeople residing in Cyprus	Health or humanitarian professionals residing in Cyprus
**Protocol**	Interview guide with semi-structured questions			One-on-one interviews using a sample of questions	Individual reviews of draft instrument based on a sample of questions
**Questions**	Opinions/attitudes/feelings regarding dimensions of occupations, occupational meaning, and tool content and format	Opinions/attitudes/feelings regarding dimensions of occupations, occupational meaning, and tool content and format	Occupational profile and history (what occupations they participate in/used to participate in)/Opinions/attitudes/feelings regarding dimensions of occupations and occupational meaning/ Opinions about tool usability, understandability, and relativity	Opinions about tool usability and understandability	Opinions about tool usability, understandability, and relativity

^1^ European University of Cyprus.

**Table 2 ijerph-20-06826-t002:** Participants’ Demographics.

	Focus Group 1	*n* = 8	Focus Group 2	*n* = 8	Review 1	*n* = 4	Review 2	*n* = 4	Focus Group 3	*n* = 5
**Role/Status**	OT professional	6	OΤ professional	2	Layperson	4	Social Worker	1	Refugee	2
	OT student	2	Researcher	2			Psychologist	1	Refugee with subsidiary protection	1
			Clinician	4			Safety officer	1	Asylum seeker	2
							Social Advisor	1		
**Age**	20–50 years old		25–50 years old	20–50 years old		20–55 years old		19–35 years old		
**Gender**	Male	2	Male	2	Male	2	Male	2	Male	3
	Female	6	Female	6	Female	2	Female	2	Female	2
**Experience** **In this field**	<6 months	2	<6 months	0			<6 months	0		
	7–12 months	3	7–12 months	2			7–12 months	0		
	13–24 months	2	13–24 months	0			13–24 months	2		
	>25 months	1	>25 months	6			>25 months	2		
**Setting**									Reception Center	2
									Shelter	1
									Semi-independent houses	1
									Community Center	1
**Country of origin**	Cyprus	4	Cyprus	2	Cyprus	3	Cyprus	4	Cameroon	1
	Greece	4	Greece	1	Greece	1			Somalia	2
			Belgium	1					Central African Republic	1
			Germany	1					Guinea	1
			Canada	1						
			Australia	1						
			USA	1						

**Table 3 ijerph-20-06826-t003:** Participants’ feedback and resulting modifications of the tool.

		Feedback	Modifications
**Focus Group 1**	**Formation/structure**	Some participants mentioned that it is difficult to clearly differentiate between categories of occupations.	Tool items (occupations) were reorganized.
		Data collection in Demographics should be abbreviated.	Demographics was converted into table format.
		The tool needs to give the opportunity to summarize the respondent’s past occupational experiences and history, as, this way, we can gather information about their skills and motives.	An occupational profile, which drew mostly from the Occupational Profile (AOTA, 2020) and was adapted to gather information that describes occupational history and experiences before arrival in the host country, was created and added.
		Some participants mentioned that the organization and structure of tool elements are not clear.	The tool format was modified to include an Introduction, Part 1 (Demographics), Part 2 (Occupational profile), Instructions, and Sections A–C (Occupational items).
	**Wording**	All the participants agree to add more examples, relative to the situation of the displaced population.	An annex (Annex 1) was created containing examples of occupations that are relative to the experience of resettlement in the host country for displaced persons concerning personal hygiene and care, life at home or within the community, religion or spirituality, managing communications with public services, self-advocating, management of personal documents and accounts, following work norms and procedures, family planning, identifying interests, skills, opportunities, and recognizing systems for psychosocial support.
		Information in the Introduction and Instructions is not clear.	The Introduction and Instructions were rephrased for clarity.
		The purpose of Column C is difficult to verbalize.	A proposed phrase was added in the Instructions.
		More information is needed about the use of the initial annex in Instructions.	More detailed information about the initial annex was provided in the Instructions. It was also renamed to Annex 2.
		For understandability, relevant examples to the state of refugees need to be added to Annex 2.	One relevant example for each of the seven categories of factors was added to Annex 2.
		Some found the scientific/occupational terms confusing.	A glossary (Annex 3) was created to help clarify these terms.
	**Assessment scale**	One participant argued that it is difficult to identify the degree of satisfaction as each occupational category can involve many sub-categories of occupations with differences in satisfaction.	Column D changed from “Comments” to “Are there any specific activities for which satisfaction might be different?” to explore this.
	**Administration**	Some of the participants claimed that the tool should be administered only by occupational therapists.	The issue would be further discussed in Focus group 2, before proceeding to the final pilot tool.
		On the other hand, the tool would be useful in settings without an established occupational therapist, which is usually the case.	
		**Feedback**	**Modification**
**Focus Group 2**	**Categorization**	Setting some “priority” occupations would be useful. On the other hand, all occupational needs should be addressed.	A new table was added (Section D) that presented all occupational items. Respondents would select the three most urgent ones to address.
		Categorization of occupational items, including play, sleep, social participation, and sexual activity occupations, is debatable.	Section A (Self-care, daily life at home and in the community, health management, and sleep/rest time), Section B (social participation, free time/play, and leisure activities), and Section C (Work and education) were reorganized.
	**Formation**	All of the participants agree that the tool is time-consuming. A short version of it needs to be created.	It was added in the Instructions to administer certain sections of the tool to curtail administration when necessary.
	**Wording**	There should be the option to omit assessment of sensitive occupations due to reasons such as culture or trauma.	The “not applicable” option of Column A changed to “not assessed”.
		Few participants stated that the tool needs to be translated into different languages for a better understanding by non-English speakers.	Researchers decided to initially create the pilot tool in English, even though language is an important part of culture, as it was designed to be administered by specialists in the field that could support respondents’ participation despite difficulties in language use, given that respondents are English speakers. Later, it could be translated into other languages.
		One participant suggested to add “survival activities” in Annex 1 as they are often indicative of occupations in which displaced persons engage in the host country to support themselves.	“Survival activities” were added in Annex 1 for relativity.
	**Administration**	It is important that the tool be administered by non-occupational therapists, as well, because often an occupational therapist is not available in refugee settings.	The Instructions changed to encompass this.
		The tool should be administered by trained experts in the field who would be able to manage ethical and practical challenges during administration pertinent to culture, language, resources, or trauma.	This was included in the Introduction.
		Administration at the initial reception phase during asylum process might be challenging as, at that point, more urgent needs have to be addressed and displaced persons need to receive emergency response aid.	In the Instructions, it was added that the tool was created as a screening tool after first response procedure.
		**Feedback**	**Modification**
**Review 1**	**Wording**	A few participants found the terms “occupation” and “satisfaction” unclear.	Explanatory definitions in the Introduction were added.
		**Feedback**	**Modification**
**Review 2**	**Categorization**	The occupation “educating/informing for legal rights and obligations” needs to be divided into two distinct occupations to reflect these two diverse notions.	The occupation was divided, adding one more evaluation component (35 in total).
	**Wording**	“Dialect” needs to be added in Demographics along with “language”, as it is important when identifying specific groups and minorities.	“Dialect” was added in the Demographics part.
	**Administration**	One participant stated that the tool would be useful for adolescents in need of psychosocial support as they transition to adulthood in a strange country that is often unwelcoming.	The tool administration age changed from >18 years old to >17 years old.
		Respondents need to be fluent in the English language to avoid misinterpretations.	The phrase “be able to communicate effectively in the English language” was added in the Introduction.
		**Feedback**	**Modification**
**Focus Group 3**	**Wording**	The purpose of Column D, which addressed occupations for which satisfaction might be different even though they fall under the same occupational category, was confusing for participants. On the other hand, other issues regarding relativity, culture, or challenges during administration needed to be addressed.	Column D again changed to “Comments”, for comments or concerns by either the interviewer or the respondent that could be considered regarding issues during administration relating to language use, cross-cultural differences, or others.
		Most of the participants mistakenly understood the term “satisfaction” as the ability to engage in occupations.	Changes were made to the Introduction to refine phrasing of the term “Satisfaction”.

## Data Availability

The data presented in this study are available on request from the corresponding author. The data are not publicly available, due to privacy and ethical restrictions.

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
