# Peer review of "Development of the Refugees and Asylum Seekers Occupational Satisfaction (RASOS) Assessment Tool"

_ijerph, 2023, doi:10.3390/ijerph20196826_

Round 1
Reviewer 1 Report
This text provides a qualitative assessment of assessment tools for vocational and client centers for refugees and asylum seekers, leading to the development of the Refugee and Asylum Seekers Occupational Satisfaction Assessment Tool (RASOS). Specifically, a convenient sample included 8 professional and student occupational therapists from Greece and Cyprus, 8 international expert occupational therapists, 4 laypersons, 4 humanitarian professionals, and 5 refugees and asylum seekers. Basic qualitative content and thematic analysis covered themes related to tool modification, including categorization, construction/structure, wording, management, and evaluation scales. However, the following issues need to be addressed properly before publication. The specific suggestions are as follows:
In the Language section
Language could be more concise, reducing complex sentences. Use a more coherent and clearer presentation, especially in the introduction, data description, and conclusion sections. Some expressions need to be more precise, for instance: change "resulting in the pilot version of the tool." to "resulting in the creation of a pilot version of the tool." for better clarity. Change "Refugees and Asylum Seekers Occupational Satisfaction 20" to "Refugees and Asylum Seekers Occupational Satisfaction (RASOS)" for a clearer reference to the tool's name.
In the Materials and Methods section:
(1)Convenience sampling was used in the second part of Materials and Methods. 31 people were invited to participate. The sample size was too small.
Suggestion: increase the sample size to increase the reliability of the article and to argue whether this sample is representative.
(2)Concerning the research process, which spanned two years and two stages – the first involving establishing theoretical background and creating the initial draft tool, and the second further developing the tool – the following issues and improvement suggestions are raised:
This research mainly employed qualitative analysis, and some conclusions are subjective, lacking scientific data validation. The creation of the draft tool was based on initial literature, whose relevance couldn't be verified.
Suggestion: A mixed-methods approach of qualitative and quantitative research could be employed. Quantitative analysis of assessment data from the draft tool compared with real-world data on occupational satisfaction among refugees and asylum seekers can help refine and improve the tool based on quantitative findings, facilitating the verification of its effectiveness and scientific validity.
(3)The extended timeline from 2020 to 2022 introduces the possibility of changes in certain factors over different periods.
Suggestion: It's recommended to incorporate appropriate time points for follow-up testing within the study to better track and analyze changes.
(4)The research method predominantly focused on focus groups and reviews, primarily using semi-structured interviews, which were limited by differences in understanding and expression due to interviewees' diverse cultural backgrounds.
Suggestion: Consider incorporating field observations to gather more comprehensive data. Observations in actual settings can provide insights into how respondents utilize the tool within real contexts, enhancing understanding of its effectiveness.
In the data analysis section
while it seems fairly rigorous and comprehensive, there are still avenues for improvement to enhance the credibility and depth of the analysis.
(1)the authors reflect that personal biases could influence the interpretation of the research and data.
Suggestion: While reporting the data analysis process, consider adding a section describing how the authors acknowledged and addressed potential biases, reducing subjective biases of the first, second, and third authors.
(2)engaging in multiple rounds of open coding can enhance the credibility of the analysis. Inviting more researchers to code the data and then comparing coding results can reveal different perspectives and patterns.
(3)In addition to cross-verifying audio transcriptions and written notes, consider conducting inter-coder validation during the analysis phase. Relying solely on mutual reviews among three authors has limitations.
Suggestion: It's suggested to have different researchers analyze the same segment of data and then compare results to confirm analysis consistency, thereby boosting credibility and depth of analysis.
In the data content
(1)the sample primarily represents resettlement in a Western high-income country (Cyprus) and possesses English proficiency. Participation of refugees and asylum seekers was limited to Focus Group 3, initially required to establish the tool's scientific basis. Unfortunately, only displaced individuals from the African continent ultimately participated, leading to data variations due to cultural background differences, social status, and economic disparities. Moreover, employing convenience sampling and surveying 31 individuals appears insufficient in terms of tool applicability.
Suggestion: To enhance applicability, diversify the sample as much as possible. Collaboration with refugees and asylum seekers from other countries or regions can provide a broader participant pool. This aids in minimizing biases towards specific regions or groups.
(2)Focus Group 2 and Focus Group 3 were conducted in English. Participants whose first language isn't English might encounter difficulties in understanding the moderator's questions, comments from other participants, expressing their views, and contributing their insights.
Suggestion: Consider conducting focus groups and reviews in multiple languages to accommodate non-English fluent participants. This can be achieved by providing appropriate translation and explanation to ensure all participants comprehend questions fully and express their viewpoints. The moderator's questioning style as elected by Focus Groups might lead to varying understandings among individuals from different cultural backgrounds. This necessitates training moderators and researchers to enhance their capabilities in cross-cultural communication and multilingual environments, fostering effective communication among participants from diverse cultural backgrounds.
In the results and discussion section:
(1)This article mentions that the initial instrument outline was based on 12 literature reviews, taking into account the strengths and limitations of previously published instruments. However, this study did not compare the strengths and weaknesses with previous literature and previously published tools in the results and discussion section.
Suggestion: in the results and discussion section, write how the results of this paper are different or the same as previous studies. What is the innovation of the tool in this paper? Why is this paper's tool better suited to this paper's research audience (refugees and asylum seekers)?
This text provides a qualitative assessment of assessment tools for vocational and client centers for refugees and asylum seekers, leading to the development of the Refugee and Asylum Seekers Occupational Satisfaction Assessment Tool (RASOS). Specifically, a convenient sample included 8 professional and student occupational therapists from Greece and Cyprus, 8 international expert occupational therapists, 4 laypersons, 4 humanitarian professionals, and 5 refugees and asylum seekers. Basic qualitative content and thematic analysis covered themes related to tool modification, including categorization, construction/structure, wording, management, and evaluation scales. However, the following issues need to be addressed properly before publication. The specific suggestions are as follows:
In the Language section
Language could be more concise, reducing complex sentences. Use a more coherent and clearer presentation, especially in the introduction, data description, and conclusion sections. Some expressions need to be more precise, for instance: change "resulting in the pilot version of the tool." to "resulting in the creation of a pilot version of the tool." for better clarity. Change "Refugees and Asylum Seekers Occupational Satisfaction 20" to "Refugees and Asylum Seekers Occupational Satisfaction (RASOS)" for a clearer reference to the tool's name.
In the Materials and Methods section:
(1)Convenience sampling was used in the second part of Materials and Methods. 31 people were invited to participate. The sample size was too small.
Suggestion: increase the sample size to increase the reliability of the article and to argue whether this sample is representative.
(2)Concerning the research process, which spanned two years and two stages – the first involving establishing theoretical background and creating the initial draft tool, and the second further developing the tool – the following issues and improvement suggestions are raised:
This research mainly employed qualitative analysis, and some conclusions are subjective, lacking scientific data validation. The creation of the draft tool was based on initial literature, whose relevance couldn't be verified.
Suggestion: A mixed-methods approach of qualitative and quantitative research could be employed. Quantitative analysis of assessment data from the draft tool compared with real-world data on occupational satisfaction among refugees and asylum seekers can help refine and improve the tool based on quantitative findings, facilitating the verification of its effectiveness and scientific validity.
(3)The extended timeline from 2020 to 2022 introduces the possibility of changes in certain factors over different periods.
Suggestion: It's recommended to incorporate appropriate time points for follow-up testing within the study to better track and analyze changes.
(4)The research method predominantly focused on focus groups and reviews, primarily using semi-structured interviews, which were limited by differences in understanding and expression due to interviewees' diverse cultural backgrounds.
Suggestion: Consider incorporating field observations to gather more comprehensive data. Observations in actual settings can provide insights into how respondents utilize the tool within real contexts, enhancing understanding of its effectiveness.
In the data analysis section
while it seems fairly rigorous and comprehensive, there are still avenues for improvement to enhance the credibility and depth of the analysis.
(1)the authors reflect that personal biases could influence the interpretation of the research and data.
Suggestion: While reporting the data analysis process, consider adding a section describing how the authors acknowledged and addressed potential biases, reducing subjective biases of the first, second, and third authors.
(2)engaging in multiple rounds of open coding can enhance the credibility of the analysis. Inviting more researchers to code the data and then comparing coding results can reveal different perspectives and patterns.
(3)In addition to cross-verifying audio transcriptions and written notes, consider conducting inter-coder validation during the analysis phase. Relying solely on mutual reviews among three authors has limitations.
Suggestion: It's suggested to have different researchers analyze the same segment of data and then compare results to confirm analysis consistency, thereby boosting credibility and depth of analysis.
In the data content
(1)the sample primarily represents resettlement in a Western high-income country (Cyprus) and possesses English proficiency. Participation of refugees and asylum seekers was limited to Focus Group 3, initially required to establish the tool's scientific basis. Unfortunately, only displaced individuals from the African continent ultimately participated, leading to data variations due to cultural background differences, social status, and economic disparities. Moreover, employing convenience sampling and surveying 31 individuals appears insufficient in terms of tool applicability.
Suggestion: To enhance applicability, diversify the sample as much as possible. Collaboration with refugees and asylum seekers from other countries or regions can provide a broader participant pool. This aids in minimizing biases towards specific regions or groups.
(2)Focus Group 2 and Focus Group 3 were conducted in English. Participants whose first language isn't English might encounter difficulties in understanding the moderator's questions, comments from other participants, expressing their views, and contributing their insights.
Suggestion: Consider conducting focus groups and reviews in multiple languages to accommodate non-English fluent participants. This can be achieved by providing appropriate translation and explanation to ensure all participants comprehend questions fully and express their viewpoints. The moderator's questioning style as elected by Focus Groups might lead to varying understandings among individuals from different cultural backgrounds. This necessitates training moderators and researchers to enhance their capabilities in cross-cultural communication and multilingual environments, fostering effective communication among participants from diverse cultural backgrounds.
In the results and discussion section:
(1)This article mentions that the initial instrument outline was based on 12 literature reviews, taking into account the strengths and limitations of previously published instruments. However, this study did not compare the strengths and weaknesses with previous literature and previously published tools in the results and discussion section.
Suggestion: in the results and discussion section, write how the results of this paper are different or the same as previous studies. What is the innovation of the tool in this paper? Why is this paper's tool better suited to this paper's research audience (refugees and asylum seekers)?
Reviewer 2 Report
This study has developed an evaluation tool of occupational satisfaction for refugees and asylum seekers, which has certain practical significance, but there is still room for further improvement. The following comments are for reference only.
1. This study lacks relevant theoretical support, and there is no basis for the selection of many indicators, so it is suggested that the author further supplement it.
2. Refugees and asylum seekers may come from different countries or regions, and their occupations may vary greatly. Did the author take this into account when designing the assessment tool?
3. The literature review is not comprehensive enough, and the author needs to cite the authoritative literature at present.
4. In the conclusion and discussion section, the author needs to further explain the value of this study and its significance to global development.
Reviewer 3 Report
Thank you for this opportunity to review the article “Development of the Refugees and Asylum Seekers Occupational Satisfaction (RASOS) assessment tool”
Introduction:
The authors provided a brief introduction about asylum seekers and occupational injustice.
I would encourage the authors delve deeper into occupational injustice, particularly exploring its effects on refugees and asylum seekers. This will provide a comprehensive understanding of the topic.
The authors also need to reorganize introduction and clarify the gaps in the current literature and assessment tools. Specifically, one paragraph that starts with, "what can be defined as a meaningful occupation is also up for debate…" is ambiguous. Could the authors clarify the intended message?
Then the authors reviewed non-occupation-specific assessment tools. I do not understand why it is necessary to review non-occupation-specific instruments. These instruments are not related to the study topic.
After that, the authors reviewed occupation-centered tools. A more in-depth exploration of these tools would enhance the paper's value. For example, understanding the constructs these tools assess and the type of results the tools can generate would be beneficial. Although these tools are not specifically designed for refugees and asylum seekers, I believe there are some good potential constructs within these tools that could be applied for assessing occupational performance for asylum seekers.
Towards the end, there's a list of assessment tools with highlighted limitations. I guess the authors intended to use the limitations as the identified gaps. I would suggest that the authors conduct a broader literature review. They should systematically organize the contributions of existing instruments and present the overarching limitations that this study seeks to address. This approach will provide a more grounded rationale for this study.
Methods:
It is good to see the authors invited participants from diverse backgrounds to review the draft tool. I would encourage the authors to explain specifically the function of focus groups and reviews, respectively. Additionally, it would be beneficial to provide a detailed table summarizing the methodologies employed for both the focus groups and reviews. The specifics should address questions like:
Were the focus groups/reviews conducted online?
How were the focus groups/reviews conducted?
What were the exact protocols and sample questions utilized during the focus groups and reviews?
Did the authors use a single protocol or were there different ones for the focus groups and reviews?
The authors also need to briefly explain how the protocol(s) was(were) developed.
For Tables 1 and 2, please consider a revision as the current format isn't reader-friendly.
Results and discussion:
For the results, I found it hard to interpret the results because the authors used many
abbreviation, such as OTPF, COPM, WHOQOL, and MOHOST. Please provide explanations for each abbreviation.
I noticed that some details presented in the discussion would be more appropriately introduced in the introduction. Specifically, the statement "Literature review demonstrates that there is a broad range of occupations that displaced persons are excluded from, thus limiting occupational satisfaction" should be incorporated into the introduction. This would provide a deeper context on the topic of occupational injustice, which I've previously mentioned.
Author Response
Please see the attatchment

Round 2
Reviewer 1 Report
Accept in present form
Accept in present form
Reviewer 3 Report
The authors made efforts to address all the comments I provided in the previous review report. I agree that the paper can be published.